# Lithium intercalated FeSe as a high-temperature superconducting ferromagnet

Yi Hu[1,13], Keyi Liang[2,13], Jie Li[3,13], Zhijie Li[2], Fanyu Meng[4,5], Hechang Lei [4,5], Jiyuan Wang[1], Huizhen Wen[1], Ruozhou Zhang[2], Jiaqiang Cai[6], Jinglei Zhang[6], Yi Lu [3,7] ✉, Yihua Wang [2,8] ✉, Qi-Kun Xue [1,9,10] ✉ & Ding Zhang [1,9,11,12] ✉

Merging superconductivity and ferromagnetism in a single material may promise unparalleled quantum properties for next-generation devices. Here, we bring together the two antagonistic phenomena at a record-high temperature via electric-field controlled lithiation of FeSe. The in-situ gating allows us to switch the simple compound of FeSe between a nonmagnetic superconductor and a superconducting ferromagnet. In the latter state, itinerant ferromagnetism persists from above 200 K to a temperature well below the superconducting transition temperature (45 K), as demonstrated not only by magneto-transport but also via scanning superconducting quantum interference device (sSQUID) microscopy. Interestingly, applying certain in-plane magnetic fields enhances superconductivity, reflecting the intimate interplay between high-temperature superconductivity and ferromagnetism. Density-functional theory calculations further reveal the instability of FeSe toward ferromagnetism at a moderate lithium concentration. These findings open up fresh opportunities in iron-based superconductors that interface dissipationless electronics and spintronics.

Harmonizing superconductivity and long-range ferromagnetism in a single material is not only of fundamental interest in extending our understanding on distinct pairing mechanisms but also promises rich applications in dissipationless electronics and spintronics[1-3]. This task is challenging because superconductivity occurs predominantly with singlet Cooper pairing, which is incompatible with ferromagnetism that requires spins to be polarized to the same direction. So far, the combination of superconductivity and ferromagnetism has been achieved only in a handful of uniform materials. For instance, U-based superconductors—UGe$_2$[4], URhGe[5], UCoGe[6] and lately UTe$_2$[7,8]—have been reported to host both superconductivity and ferromagnetic interactions, possibly resulting in the intriguing spin triplet Cooper pairs. Still, their superconducting transition temperature $T_{sc}$ only lingers around 1 K and the corresponding magnetic temperature $T_m$ stays below 30 K.

One possible route to synthesizing a material with high-$T_{sc}$ and high-$T_m$ is by stacking cuprate or iron-based superconductors with ferromagnetic layers[3,9-13]. For instance, in EuFe$_2$(As$_{1-x}$P$_x$)$_2$[11] and [(Li,Fe)

[1]State Key Laboratory of Low Dimensional Quantum Physics and Department of Physics, Tsinghua University, Beijing, China. [2]State Key Laboratory of Surface Physics and Department of Physics, Fudan University, Shanghai, China. [3]National Laboratory of Solid State Microstructures and Department of Physics, Nanjing University, Nanjing, China. [4]School of Physics and Beijing Key Laboratory of Optoelectronic Functional Materials & MicroNano Devices, Renmin University of China, Beijing, China. [5]Key Laboratory of Quantum State Construction and Manipulation (Ministry of Education), Renmin University of China, Beijing, China. [6]Anhui Province Key Laboratory of Condensed Matter Physics at Extreme Conditions, High Magnetic Field Laboratory, HFIPS, Chinese Academy of Sciences, Hefei, Anhui, China. [7]Collaborative Innovation Center of Advanced Microstructures, Nanjing University, Nanjing, China. [8]Shanghai Research Center for Quantum Sciences, Shanghai, China. [9]Beijing Academy of Quantum Information Sciences, Beijing, China. [10]Southern University of Science and Technology, Shenzhen, China. [11]Hefei National Laboratory, Hefei, China. [12]RIKEN Center for Emergent Matter Science (CEMS), Wako, Saitama, Japan. [13]These authors contributed equally: Yi Hu, Keyi Liang, Jie Li. ✉e-mail: yilu@nju.edu.cn; wangyhv@fudan.edu.cn; qkxue@mail.tsinghua.edu.cn; dingzhang@mail.tsinghua.edu.cn

OH]FeSe[12,13], the superconducting layers of FeAs ($T_{sc}$ = 26 K) and FeSe ($T_{sc}$ = 43 K) are sandwiched by ferromagnetic layers of Eu ($T_m$ = 20 K)[11] and the blocks of LiFeOH hosting canted antiferromagnetism ($T_m \sim$ 10 K)[12,13]. There, however, the onset for magnetism stays low and the coupling between superconducting and magnetic layers remains limited due to their obvious vertical displacement. Recently, in [(Li,Fe) OH]FeSe, a ferromagnetic state with the Curie temperature higher than 175 K was realized after the full suppression of superconductivity by lithium intercalation[14]. It has prompted further theoretical investigations on a simpler compound−FeSe, leading to the proposal of room-temperature quantum anomalous Hall effect in LiFeSe[15]. In practice, studies on Li-intercalated FeSe largely focused on the enhanced superconductivity ($T_{sc} \sim$ 40 K)[16–18]. Indication of anomalous Hall effect was recently observed at 50 K[18], calling for a comprehensive study of possible ferromagnetism in Li$_x$FeSe with x∼50%. Whether or not the ferromagnetism coexists with superconductivity in lithium intercalated FeSe remains an open problem.

Here, by combining in-situ solid-state gating with magneto-transport and magnetic imaging, we unambiguously establish lithium intercalated FeSe as a superconducting ferromagnet with record-high temperatures of both $T_{sc}$ and $T_m$. In multiple samples with $T_{sc}$ = 45 K, we observe pronounced hysteresis loops in both Hall and longitudinal resistances up to 225 K, attesting to itinerant ferromagnetism with $T_m$ > 225 K. The electric-field controlled gating allows us to switch the system reversibly between the nonmagnetic superconducting state and the superconducting ferromagnetic state. Scanning super-conducting quantum interference device (sSQUID) magnetometry and susceptometry reveal that the ferromagnetic domains survive deep in the Meissner state and they coexist spatially. Moreover, we unveil that superconductivity gets enhanced at intermediate in-plane magnetic

fields (4–12 T), before reverting to conventional response at higher magnetic fields (measured up to 35 T). This unusual non-monotonic response suggests possible involvement of spin-polarized Cooper pairs, stemming from the strong interplay between ferromagnetism and superconductivity. Our calculations based on density functional theory further identify that ferromagnetism occurs at around 50% doping of lithium, i.e., Li$_{0.5}$FeSe. These findings have profound implications on the pairing mechanism of iron-based superconductors and establish a distinct route that combines high-$T_{sc}$ superconducting electronics and high-$T_m$ spintronics.

## Results

### High-temperature superconductivity and ferromagnetism

Figure 1 demonstrates the electric switching between the low-$T_{sc}$ nonmagnetic phase and the high-$T_{sc}$/$T_m$ phase. Lithium intercalation into FeSe is realized by using a solid-state ion backgate[16–21] (Fig. 1a). An exfoliated thin flake of FeSe capped by the hexagonal boron nitride (h-BN), as illustrated in Fig. 1b for sample S1 (15 nm thick), is positioned on the solid ion conductors (SIC) and contacted by the bottom electrodes (details in Methods). The pristine sample has $T_{sc,0}$ = 5.2 K (Fig. 1c), which is lower than that of bulk FeSe due to reduced thickness[22] (We define $T_{sc,0}$ as the temperature point above which the zero-field resistance reaches 1% of the normal state resistance). It shows a linear Hall effect with a positive slope (Fig. 1d), suggesting hole-dominant transport. Meanwhile, the magneto-resistance $MR = (R_{xx}(B) − R_{xx}(0))/R_{xx}(0)$ shows a slight increase with the increasing magnetic field (Fig. 1e). We then apply a positive backgate voltage at temperatures around 260 K to intercalate lithium ions into FeSe. The intercalation stops by cooling the sample below 200 K. By optimizing the gating conditions (voltage and timespan), we realize a rather

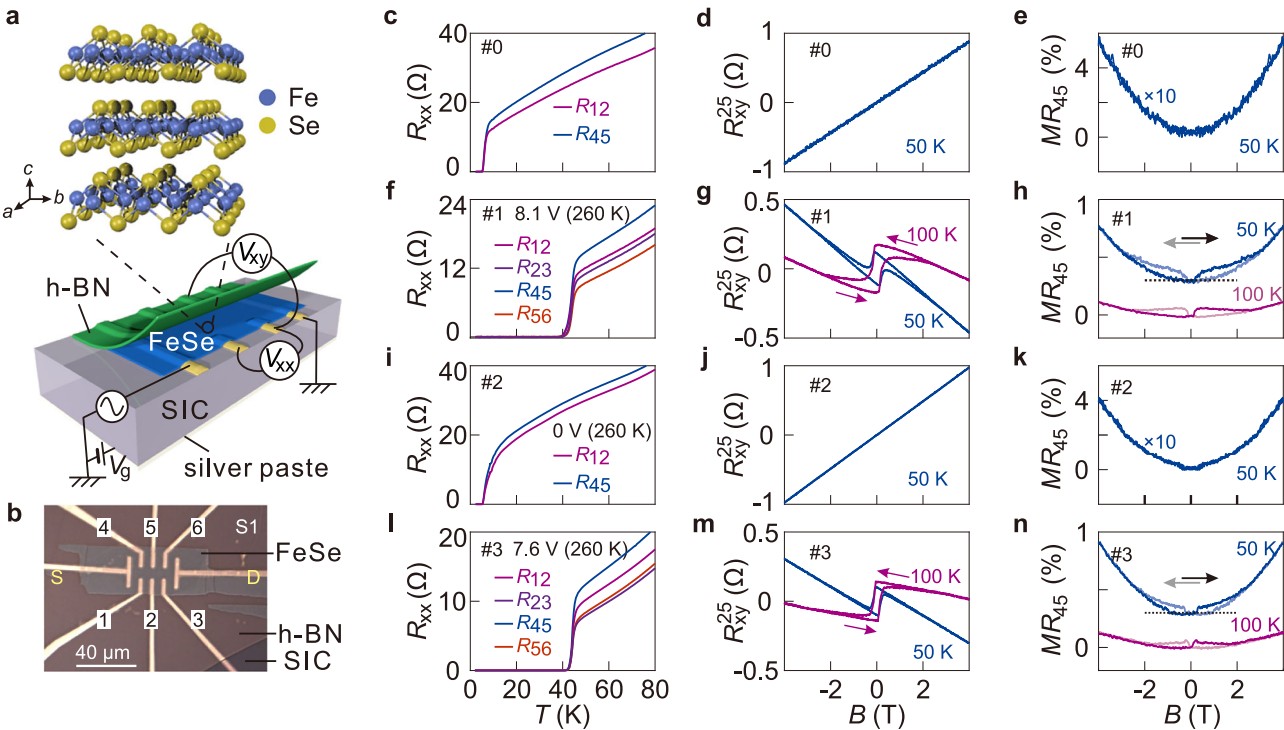

**Fig. 1 | Switching between a nonmagnetic superconductor and a high-temperature superconducting ferromagnet. a** Schematic drawing of the experimental setup illustrating an exfoliated flake of FeSe on the solid ion conductor (SIC) and capped by h-BN. **b** Optical image of sample S1 (about 15 nm thick). Numbers indicate the electrodes used for measuring the resistance. Letters S and D indicate the source and the drain. **c-n** Temperature dependent resistance, Hall resistance, and magnetoresistance (MR) collected from S1 in four different states:

#0, the pristine state (**c–e**); #1, lithium intercalated state (**f–h**); #2, de-intercalated state (**i–k**); #3, re-intercalated state (**l, m, n**). Voltage and temperature values in **f, i, l** represent the gating conditions. Sub/superscripts of R with numbers indicate the corresponding pair of contacts (shown in Fig. 1b). Hall and magnetoresistance data are anti-symmetrized from the raw data (see Methods). Arrows indicate the sweeping direction of the magnetic field. Curves in **h** and **n** are vertically offset for clarity (Dotted lines mark zero for the offset curves).

uniform state on the mesoscopic scale, as reflected by the traces measured from different pairs of contacts (Fig. 1f). Lithium intercalation promotes the onset transition temperature $T_{sc}$ to a value of 45 K ($T_{sc,0}$ = 39 K), in agreement with previous studies[16–18]. The negative slope in the ordinary Hall effect (Fig. 1g) also confirms that the transport is dominated by electrons. We estimate the lithium content to be about 48%, based on the change in the Hall densities at 50 K (Supplementary note 3). This lithium concentration for the high-$T_{sc}$ phase is consistent with previous studies[17].

Notably, we observe pronounced hysteresis loops in both the Hall (Fig. 1g) and the *MR* (Fig. 1h) in the high-$T_{sc}$ phase. Similar hysteresis loops can be obtained from different pairs of contacts (Supplementary Fig. 4). On the higher temperature side, the hysteretic behavior persists to a temperature as high as 200 K (Supplementary Fig. 5a). Further increasing the temperature results in activation of the lithium intercalation/de-intercalation process such that the field sweep does not show a closed loop. Nevertheless, we see jumps in the Hall resistance at the coercive fields even up to 225 K (Supplementary Fig. 5b), suggesting that $T_m > 225$ K.

The de-intercalation process (at 260 K and zero gate voltage) eventually leads to the recovery of the original low-$T_{sc}$ phase in the same sample, as demonstrated in Fig. 1i–k. As $T_{sc,0}$ returns to be around 5 K, the Hall effect (Fig. 1j) resumes the linear response while the *MR* shows a simple parabolic response (Fig. 1k). In Fig. 1l–n, we demonstrate the repeatability of this electric-field controlled switching by intercalating lithium again into the sample. Interestingly, this gated state #3 has a slightly higher $T_{sc,0}$ of 42 K. The electron density is also higher, as indicated by the shallower Hall slopes in Fig. 1m than those of Fig. 1g at the same temperatures. It indicates that state #3 has a slightly higher lithium content than state #1 does. Accompanied by this further increase in lithium concentration, the coercive field becomes smaller (Supplementary Fig. 6)[18]. The implication of these results will be given in the discussion section.

### Influence of superconductivity on ferromagnetism

Figure 2a–d show a systematic evolution of the itinerant ferromagnetism around the superconducting transition. The results are collected from a second sample−S2 (14 nm thick)−in the high-$T_{sc}$ phase (Fig. 2a), demonstrating the reproducibility of the observed phenomena. Similar results from other pairs of contacts of S2 are given in Supplementary Fig. 7. The anomalous Hall resistance shows an apparent sign reversal around $T_{sc}$, as demonstrated in Fig. 2b (Supplementary Fig. 8). By contrast, the ordinary Hall component shows little variation throughout the superconducting transition, as shown by the extracted Hall density (diamonds) as a function of temperature in Fig. 2d. It corresponds to a fixed Fermi level thus little change in the Berry curvature. Therefore, the anomalous Hall component from the non-trivial Berry curvature[23] stays unchanged throughout the superconducting transition. Apart from this intrinsic mechanism, skew scattering can contribute to the AHE[23]. Indeed, the positive $R_{AH}$ scales linearly with the longitudinal resistance $R_{xx}$ in the temperature window from 180 to 50 K (Supplementary Fig. 9), supporting the presence of skew scattering. The condensation of Cooper pairs dramatically suppresses scattering such that the intrinsic mechanism dominates at low temperatures. This two-component process leads to a sign reversal of $R_{AH}$. Interestingly, there exist peak/dip structures in the Hall traces around the coercive field at 45 to 42 K (Fig. 2b), reminiscent to the transport signature of topological Hall effect (THE)[24,25]. This feature may also reflect the competition between two anomalous Hall components[26,27].

### Persistence of ferromagnetism in the superconducting state

At temperatures below 42 K, corresponding to $T_{sc,0}$ of S2, the anomalous Hall effect gets strongly suppressed but the magnetic field dependence of the longitudinal resistance $R_{xx}(B)$ still shows the butterfly pattern (Fig. 2c). Even at 35 K, well below $T_{sc,0}$, clear hysteresis can be observed (Supplementary Fig. 10). We exclude the magnetocaloric effect here by measuring the hysteresis at a sufficiently low sweeping rate (Supplementary Fig. 10). We also comment that $R_{xx}(B)$ measured in the superconducting transition regime shows dips around the coercive field. This is opposite to those observed at higher temperatures (Fig. 1h and n). It indicates that the randomization of spin polarization reduces flux lines penetrating through the sample, effectively enhancing transport of Cooper pairs in the superconducting regime.

In order to investigate the spatial coexistence of superconductivity and ferromagnetism, we employ sSQUID microscopy on lithium intercalated FeSe (sample S3, about 14 nm thick). This technique allows simultaneous imaging of local AC susceptibility (susceptometry) and static magnetic fields (magnetometry)[28–30]. The data obtained by susceptometry (Fig. 2e) reveals clear diamagnetic signal at 16 K, which persists to about 28 K. Such a signal vanishes at 34 K, which is above $T_{sc,0}$ of this sample (33 K). This temperature dependence is further summarized in Fig. 2f. The susceptometry image at 16 K indicates that nearly the entire sample, within the boundary (demarcated as dashed lines in Fig. 2e), has strong Meissner diamagnetism. Importantly, the magnetometry image of the same area (Fig. 2g) shows strong flux contrast of both positive and negative field with respect to the zero outside the sample. This is noticeably different from the vortex patterns of superconductors[31]. Instead, the patterns are similar to those from ferromagnetic domains as observed in two-dimensional ferromagnets $CrBr_3$[32] as well as the ferromagnet-superconductor heterostructure of $EuO/KTaO_3$[33]. The magnetometry pattern corresponds to the spontaneously emerged ferromagnetic domains in sample S3 in zero-field cooling. Its weak temperature dependence (Fig. 2h) is expected, because $T_m$ is much higher than $T_{sc}$ here. We remark that such a ferromagnetic pattern fully vanishes after taking a de-intercalation process on the same sample (Supplementary Fig. 15), confirming the intimate link between ferromagnetism and lithium doping. Overall, the magnetometry and susceptometry images (Fig. 2e–h) clearly demonstrate that ferromagnetism and superconductivity coexist on a mesoscopic scale within our resolution (around 2 μm).

### Easy-axis of ferromagnetism

We further study the angular dependence of ferromagnetism. Figure 3a shows the Hall effect at several angles between the magnetic field and the sample plane at 50 K (Supplementary Fig. 11a shows the corresponding *MR* data). As the magnetic field tilts to the in-plane direction, $R_{AH}$ becomes suppressed and the coercive field becomes larger, indicating that the ferromagnetic easy-axis aligns with the *c*-axis of FeSe. We remark that there exists a slight misalignment in θ of about 0.03 degree. It explains the sign-reversal in the chirality of the hysteresis loop from that at the nominal angle of 0.1° to that at 0° (indicated by the arrows on the bottom two curves in Fig. 3a). We summarize the angular dependence of the Hall coefficient (Supplementary Fig. 11c). The Hall signal follows sin(θ) trend, reflecting the two-dimensional transport behavior of the FeSe thin flakes.

### Enhanced superconductivity in an in-plane magnetic field

After aligning the magnetic field in the plane, we study the superconducting transition at different field strengths. Figure 3b plots the data of S2. Applying a magnetic field usually weakens superconductivity such that resistance traces at higher *B* monotonically shift to lower temperatures. This standard behavior is obviously violated here. As indicated by the red arrow in Fig. 3b, a large portion of the resistance curve shifts to higher temperatures at higher values of *B*, leaving only a small section that shifts to lower temperatures (gray arrow). Similar behavior can be observed by using other pairs of contacts (Supplementary Fig. 13). Such a dichotomy gives rise to a fixed

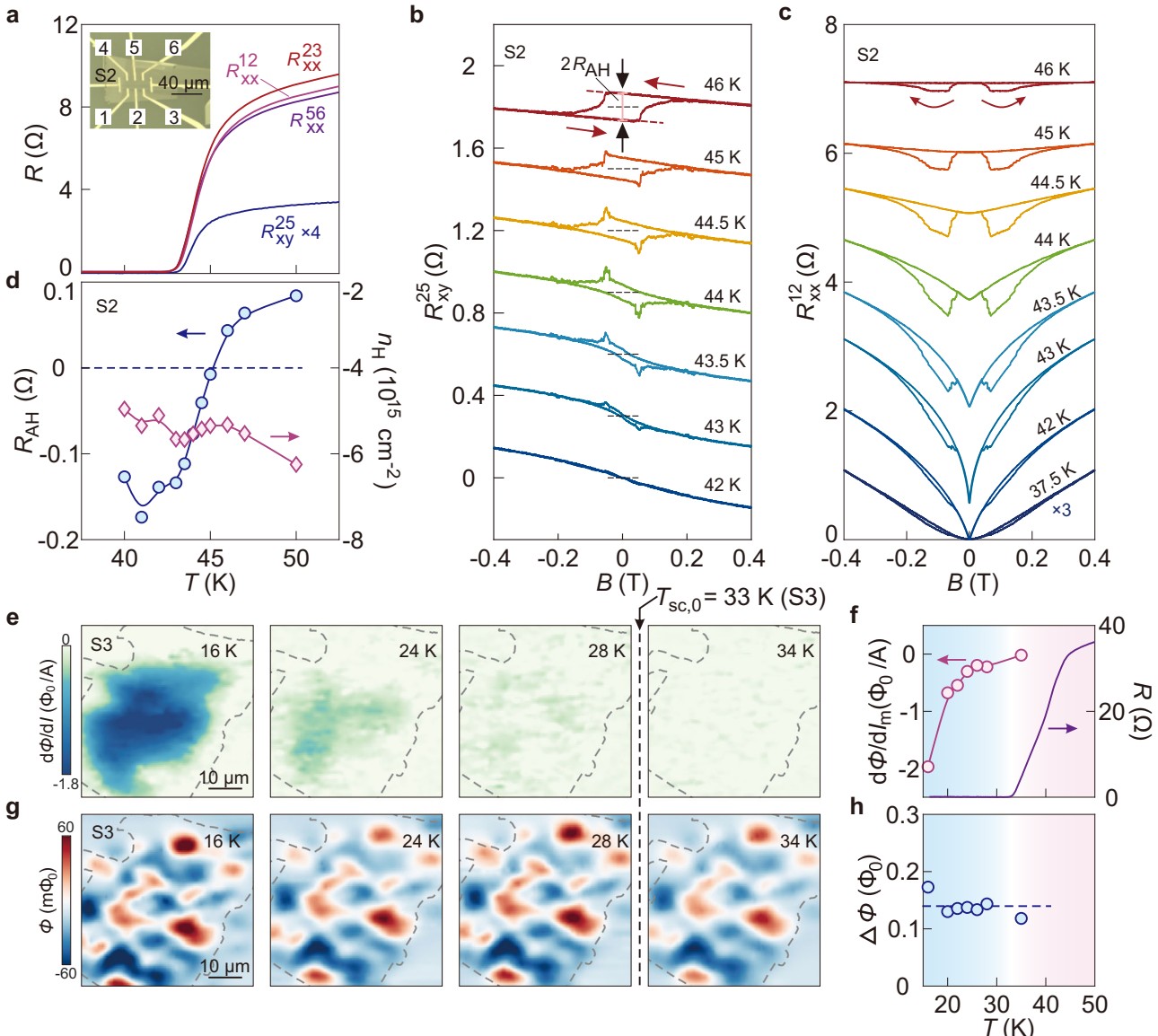

**Fig. 2 | Coexistence of superconductivity and ferromagnetism in lithium intercalated FeSe. a** Temperature dependent resistance measured from different pairs of contacts on sample S2 after lithium intercalation. Inset is an optical image of sample S2, taken before capping with h-BN. Upper indices of $R_{xx}$ or $R_{xy}$ correspond to the contacts shown in the inset. **b** Hall resistance as a function of perpendicular magnetic field measured at a set of temperature points. Curves are vertically offset (dashed lines indicate zero Hall resistances). Black arrows indicate the definition of the anomalous Hall resistance $R_{AH}$. **c** Longitudinal resistance as a function of perpendicular magnetic field at a set of temperature points. Arrows denote the sweeping directions. **d** Temperature dependence of $R_{AH}$ (circles) and the Hall carrier density (diamonds). **e** Susceptometry images measured at a set of temperature points on sample S3. Dashed curves delineate the sample boundary. **f** Temperature dependence of the maximum diamagnetic strength (circles) and sample resistance (solid curve) at zero field. **g** Magnetometry images measured at a set of temperature points. Dashed curves delineate the sample boundary. **h** Temperature dependent magnetic contrast $\Delta\Phi = \Phi_{max} - \Phi_{min}$. $\Phi_{max}$ and $\Phi_{min}$ are the maximum and minimum magnetic flux values in the scanned area, respectively.

point (marked by the dashed line), at which the resistance stays constant with $B$. We reason that this unusual behavior is caused by the convolution of two effects (inset of Fig. 3b). First, there exists a shift of the transition edge to higher temperatures, opposite to the standard response. It reflects enhanced superconductivity by the applied in-plane magnetic fields. Secondly, the superconducting transition broadens at higher fields, presumably due to the entrance of vortices (Supplementary Note 5). Such a broadening effect in addition to the enhanced superconductivity was reported in the study of Pb films under a tiny tilting of the magnetic field[34]. We reproduce the peculiar response to in-plane magnetic field in sample S4 (about 14 nm thick). We tune the lithium content such that this sample enters the high-$T_{sc}$ phase with $T_{sc} = 45$ K. Figure 3c shows the magneto-resistance of S4 in

the superconducting transition region. Instead of the conventional positive slope, we observe clearly negative magneto-resistance (indicated by the red arrow) up to about 15 T, indicating re-entrant superconductivity. Further increasing $B$ recovers the standard response.

The enhanced superconductivity is very sensitive to the tilting of the magnetic field. In Fig. 3d and Fig. 3e, we measure the temperature-dependent resistance at various $\theta$ and at a fixed magnetic field of 12 T. The white dashed line demarcates the superconducting transition temperature by using the mid-point criterion ($50\% R_n$), i.e., $T_{sc}^{50\%}(B = 12T)$. From this measurement, it becomes clear that $T_{sc}^{50\%}(B = 12T)$ around $\theta \leq \pm 1.5°$ surpasses the zero-field value. Out of this small range of angles, however, the superconducting transition

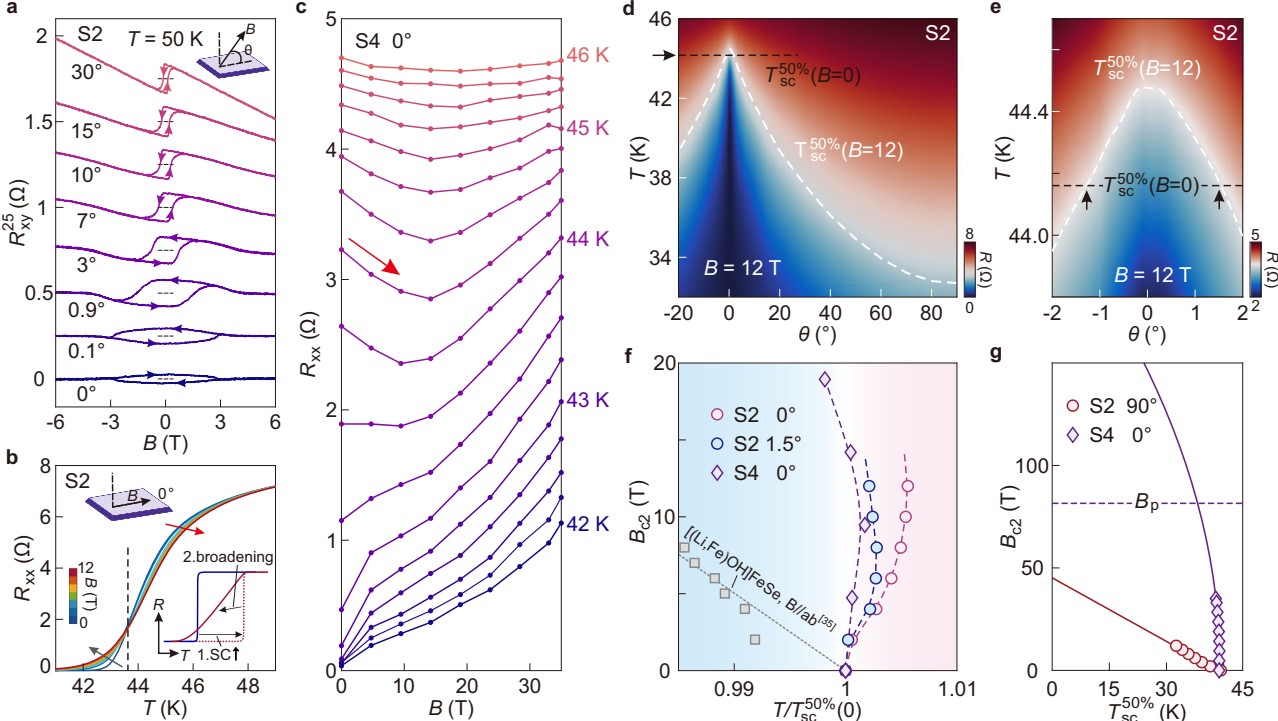

**Fig. 3 | Enhanced superconductivity under an in-plane magnetic field. a** Hall resistance as a function of magnetic field at selected tilting angles for sample S2. Inset illustrates the definition of the tilting angle. Curves are vertically offset, with the dashed lines representing zero Hall resistances. Arrows on the curves indicate the sweeping directions. The hysteresis loop at the nominal 0 degree looks clockwise, whereas other loops are counterclockwise, indicating that the real in-plane direction is between nominally 0° and 0.1°. **b** Temperature-dependent longitudinal resistances $R_{xx}$ at a set of in-plane magnetic fields for sample S2. The magnetic field increases in a step of 2 T from 0 T to 12 T. Gray and red arrows indicate the conventional and anomalous field responses, respectively. The inset illustrates two effects involved when applying an in-plane magnetic field: 1. Enhanced superconductivity (SC); 2. Increased broadening. **c** $R_{xx}$ as a function of

magnetic field at a set of temperature points for sample S4. The temperature increases in a step of 0.25 K from 42 K to 46 K. **d, e** Color-plot of $R_{xx}$ at 12 T as a function of temperature and tilting angle for sample S2. Black dashed line represents the superconducting transition temperature defined by the mid-point criterion at zero magnetic field: $T_{sc}^{50\%}(B=0)$. White dashed curve represents the transition temperature at 12 T: $T_{sc}^{50\%}(B=12)$. Arrows in **e** indicate the critical angles. Within this small range, $T_{sc}^{50\%}(B=12) > T_{sc}^{50\%}(B=0)$. **f** Upper critical fields as a function of normalized temperature at different tilting angles. Here we apply the mid-point criterion to the data from S2 and S4. Squares are the in-plane upper critical field data of [(Li,Fe)OH]FeSe[35]. **g** Upper critical fields as a function of temperature at different tilting angles. Solid lines are fits employing the 2D Ginzburg–Landau formula. Purple dashed line represents the Pauli limit.

recovers the standard response to $B$ (Supplementary Fig. 12). In Fig. 3f, we summarize the in-plane magnetic field responses for both S2 and S4. This behavior is distinctly different from the well-documented magnetic field responses of [(Li,Fe)OH]FeSe with a comparable $T_{sc}$[35]. We take the mid-point criterion to define the critical temperature at different magnetic fields. Notably, the positive shift becomes obvious only when the magnetic field reaches above 4 T−higher than the coercive field of about 3 T in the in-plane case (Fig. 3a). We further employ the two-dimensional Ginzburg-Landau formula to fit the data of S4 in the high field section (from 18.9 T to 35 T). As shown in Fig. 3g, the extrapolation indicates that the in-plane upper critical magnetic field can greatly exceed the Pauli limit[36].

### Theoretical calculations

We now employ density functional theory to evaluate the ground state energy of FeSe under lithium doping (Details in methods and Supplementary Note 1). Different magnetic states−antiferromagnetic (AFM) and ferromagnetic (FM)−are taken into considerations. In agreement with the previous study[15], it is energetically favorable to form a ferromagnetic state when the lithium content reaches 100%, corresponding to LiFeSe. Importantly, we check the tendency toward ferromagnetic coupling at intermediate lithium concentrations that are experimentally more relevant. As shown in Fig. 4a, the ferromagnetic state indeed becomes the ground state at a lithium doping that is substantially lower than 100%. The AFM-FM crossover occurs at around 50% within the range of moderate effective Coulomb

interactions ($U \in [0, 2]$ eV). This lithium content echoes with the value we experimentally estimate for the high-$T_{sc}/T_m$ phase (for instance, sample S1 in Fig. 1).

### Discussion

By now, we demonstrate that lithium intercalated FeSe−Li$_x$FeSe ($x \sim 0.5$)−shows the coexistence of superconductivity and itinerant ferromagnetism. Among the materials reporting such a coexistence (Fig. 4b), Li$_x$FeSe hosts the highest $T_{sc}$ and $T_m$. Both transport and magnetic imaging indicate a rather uniform coexistence down to the micrometer scale. On a sub-micrometer scale (10-100 nm), however, the system may still have local variation in the lithium content such that ferromagnetic puddles percolate in a superconducting matrix. Experimentally, we evaluate that the high-$T_{sc}$ phase corresponds to an average lithium content of about 50%. The distribution of lithium concentration can fluctuate around the mean value such that some local regions host a lithium content higher than 50%, crossing over to the theoretically calculated ferromagnetic phase. We remark that the sample in the high-$T_{sc}$ phase can tolerate different extent of lithium variation[17]. For the sample with a sharper transition and a higher $T_{sc,0}$ (Fig. 1l), the average lithium content is higher (as reflected by a higher Hall density) but the variation is reduced[18]. This evolution gives rise to reduction in coercive fields with increasing lithium concentration (Fig. 1g vs. Figure 1m).

So far, enhanced superconductivity under an in-plane magnetic field has been observed in various systems with relatively low $T_{sc}$,

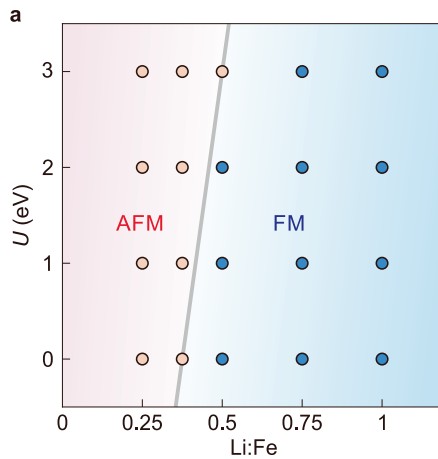

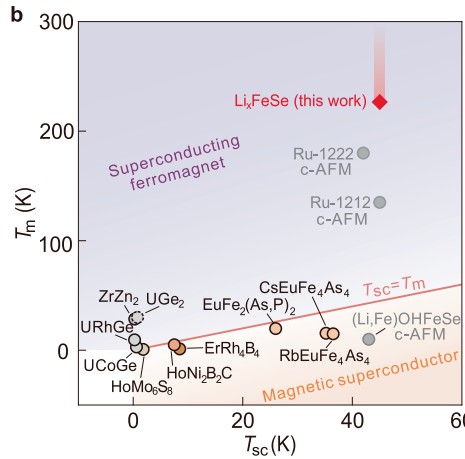

**Fig. 4 | Theoretical phase diagram and overview of superconducting ferromagnets/magnetic superconductors. a** Calculated magnetic phase diagram of FeSe at various lithium dopings and effective Hubbard U. Pink/blue symbols indicate that the calculated ground state is antiferromagnetic (AFM)/ferromagnetic (FM). **b** Overview of material systems showing coexistence of superconductivity and ferromagnetism. Shaded bar indicates that $T_m$ of $Li_xFeSe$ exceeds the highest temperature that we measure (225 K). Apart from $Li_xFeSe$ of this work, other superconducting ferromagnets ($T_m > T_{sc}$) include: cuprates (Ru-122$_2$: $Gd_{1.4}Ce_{0.6}RuSr_2Cu_2O_{10-\delta}$[9]; Ru-1212: $RuSr_2GdCu_2O_8$[10]), U-based materials ($UGe_2$ under high pressure[4], $URhGe$[5], $UCoGe$[6], $ZrZn_2$[50]). Magnetic superconductors ($T_m < T_{sc}$) include: $HoMo_6S_8$[51], $HoNi_2B_2C$[52], $ErRh_4B_4$[53], $RbEuFe_4As_4$[2], $CsEuFe_4As_4$[2], $EuFe_2(As_{0.7}P_{0.3})_2$[11], $[(Li,Fe)OH]FeSe$[12,13]. Among the listed materials, Ru-1222, Ru-1212 and $[(Li,Fe)OH]FeSe$ host weak ferromagnetism arising from canted-AFM (c-AFM) and their corresponding $T_m$-values represent the temperature for AFM ordering.

such as $Eu(Fe_{0.81}Co_{0.19})_2As_2$[37], Pb films and $LaAlO_3/SrTiO_3$ heterostructures[34], and more recently twisted bilayer/double bilayer/trilayer graphene[38–40]. For a magnetic field in the in-plane direction, its orbital pair-breaking effect is largely suppressed such that the magnetic field mainly acts on the spin degree of freedom. The underlying mechanisms involving the spins worth further discussions. For instance, materials consisting of a stack of ferromagnetic layers and superconducting layers may show enhanced $T_{sc}$ when the in-plane magnetic field rotates the spin polarization of the ferromagnetic layer, reducing its negative influence on the neighboring superconducting layer. This spin-reorientation mechanism seems unlikely to account for the observation in our case, because the enhanced superconductivity occurs in the field range where spins are already polarized to the in-plane direction (above 3 T) (Fig. 3a).

In materials hosting magnetic impurities, Jaccarino and Peter proposed that enhanced superconductivity occurs when an external magnetic field compensates for an effective internal field imposed on itinerant electrons[41]. Such a scenario demands an effective internal field stemming from the AFM coupling between the local magnetic moment−typically from rare earth spin−and the itinerant electron. In practice, the Jaccarino-Peter effect manifests itself as suppression of superconductivity at small magnetic fields (due to the alignment of local moments) that is followed by reentrant superconductivity at sufficiently high magnetic fields[42]. By contrast, we observe a continuously enhanced $T_{sc}$ as $B_\parallel$ increases from zero. We also comment that high-$T_{sc}$ superconductivity and high-$T_m$ FM in $Li_xFeSe$ stem from the same Fe 3d orbitals[15]. This situation differs from the classical scenario of Jaccarino-Peter effect, which considers exchange interaction between conduction electrons and rare earth spins. In addition, Kogan and Nakagawa proposed enhanced superconductivity in clean two-dimensional superconductors[43]. $Li_xFeSe$ seems to be in the clean limit because the estimated superconducting coherence length (based on data in Fig. 3g) is 2.7 nm while the mean free path is about 25 nm (Methods). However, the pristine FeSe flake is also in the clean limit (Methods) but no enhancement in $T_{sc}$ was observed under the in-plane magnetic field[44]. Similarly, enhanced superconductivity due to spin-orbit interactions[34] can be excluded because Li intercalation does not introduce heavy elements into FeSe.

We speculate that the enhanced superconductivity may stem from the emergence of exotic pairing−the spin-triplet pairing. The interconnection of ferromagnetic and superconducting regions in lithium intercalated FeSe is advantageous for proximity induced unconventional pairing[3]. The proximity effect is expected to be stronger than that in $EuFe_2(As_{1-x}P_x)_2$ or $[(Li,Fe)OH]FeSe$, where the ferromagnetic layer is vertically displaced from the superconducting layer. In $Li_xFeSe$, the superconducting regions can coexist with ferromagnetic puddles both laterally and vertically. The close vicinity, together with the same electronic orbitals involved[15], can help induce a spin-triplet pairing component. The existence of spin-polarized pairing component is favorable under the in-plane magnetic field. In turn, it gives rise to enhanced superconductivity. Investigating the spin structure of lithium intercalated FeSe may be an important next step toward the establishment of high-temperature spin-triplet superconductivity. In general, $Li_xFeSe$ is a unique platform that interfaces high-$T_{sc}$ superconductivity with high-$T_m$ ferromagnetism, promising transformative technology of dissipationless spintronic devices.

## Methods
### Sample growth and fabrication
Single crystals of FeSe were grown by a temperature-gradient assisted flux method[45]. High-purity Fe and Se powder were mixed in a quartz ampoule at a 1:1 ratio with a eutectic mixture of $AlCl_3$ and NaCl in the ratio of 0.52:0.48. The ampoule was evacuated down to 0.01 mbar, sealed, and placed horizontally in a furnace with a uniform temperature gradient from 620 K to 700 K for 30 days. After the growth period, we switched off the furnace and let the ampoule cool down to room temperature. After rinsing multiple times with distilled water and ethanol, FeSe crystals were dried in an oven at 100 °C for a few minutes and stored in a glovebox with Ar atmosphere.

The pristine FeSe crystals exhibited an onset superconducting transition temperature of 8 K (Supplementary Fig. 2). The Energy Dispersive X-ray (EDX) spectroscopy indicated a Fe:Se ratio of 0.92:1, suggesting a slight deficiency of iron. The crystals were mechanically exfoliated in an argon-glovebox ($H_2O < 0.1$ ppm and $O_2 < 0.1$ ppm)[19–21]. The exfoliated flakes were dry transferred onto the solid ion conductors (chemical formula: $Li_2O-Al_2O_3-SiO_2-P_2O_5-TiO_2-GeO_2$, from Ohra corp.) with prepatterned electrodes of Ti and Au (10/30 nm) by using electron beam lithography. We transferred a h-BN flake to cover FeSe for protection. Before transport measurements, the sample thicknesses of S1 and S2 were determined by atomic force microscopy

(AFM) by scanning over a corner of the FeSe flake that was not covered by h-BN. For other samples, we estimated the sample thicknesses based on the optical contrast.

## Measurement

The electrical transport measurements up to 12 T were carried out by using two closed-cycle cryogenic systems equipped with superconducting magnets. Measurements up to 35 T were carried out in the high magnetic field facility by using the water-cooled resistive magnet. We employed the standard low-frequency lock-in technique with an excitation current of 5 μA (13.33 Hz). A DC source-meter was used to apply the back-gate voltage at 300 or 260 K. To realize the high-$T_{sc}$ phase, we typically started to cool down the sample when the resistance reached to about 40% of its initial value under back-gating. The magnetic field sweep rate was 0.1 T/min for the data presented in Fig. 1 and Fig. 3a, and 0.005 T/min for the data in Fig. 2b, c.

For the angular dependent study of sample S2, we employed a home-built insert with a piezo-driven rotator (angular precision: 0.006°). The rotation axis is perpendicular to the axis of the solenoid magnet. For obtaining the data in Fig. 3b of sample S2, we monitored the sample resistance and continuously rotated the sample at a fixed magnetic field of 2 T and a fixed temperature of 40 K (in the superconducting transition regime). The in-plane direction was chosen as the angle at which the resistance showed a minimum. It utilized the strong anisotropy of the upper critical magnetic fields in Li-intercalated FeSe. We employed a similar procedure to align the magnetic field to the in-plane direction of sample S4. However, the rotator here is controlled by mechanical gears, offering an angular precision of better than 1°.

## Data processing

We presented in the main text the magnetic field dependent Hall and longitudinal resistances after anti-symmetrizing/symmetrizing the raw data to exclude the possible mixing of the two diagonal resistance components. Specifically, we took two field sweeps in the opposite directions and measured $R_{H,\rightarrow}(B)$, $R_{H,\leftarrow}(B)$, $R_{l,\rightarrow}(B)$, and $R_{l,\leftarrow}(B)$. Here, $H$ and $l$ indicate Hall and longitudinal resistances. Right/left arrows indicate the sweeping directions. We calculated $R_{xy}(B)$ and $R_{xx}(B)$ by using:

$$R_{xy}(B) = \left[R_{H,\rightarrow}(B) - R_{H,\leftarrow}(-B)\right]/2$$

$$R_{xx}(B) = \left[R_{l,\rightarrow}(B) + R_{l,\leftarrow}(-B)\right]/2$$

Both $R_{xy,xx}(B)$ and $R_{xy,xx}(-B)$ are shown in the figures. We remark that pronounced hysteresis already exists in the raw data.

## Estimation of mean free path

We employ the following formula for the evaluation of the mean free path[20]:

$$l = \frac{h/e^2}{R_{sheet}} \frac{1}{\sqrt{2\pi g n}},$$

Where $h/e^2 = 25813\ \Omega$ is the quantum resistance, $R_{sheet}$ is the sheet resistance, $g$ represents the valley degeneracy, and $n$ is the carrier density. Based on the Hall effect data in Fig. 1d and Fig. 1m, we extract the carrier density to be $2.82 \times 10^{15}\ cm^{-2}$ (holes) and $6.29 \times 10^{15}\ cm^{-2}$ (electrons) at 50 K before and after the lithium intercalation. We employ $g = 1$ for the hole band and $g = 2$ for the electron pocket of FeSe. In pristine FeSe at 50 K the transport is dominated by holes such that we input the experimentally measured sheet resistance to calculate $l$. For the lithium intercalated sample, transport is dominated by the electron bands instead. We then evaluate $l$ to be 19 nm for sample

S1 in state #0 and 25 nm in state #3. The pristine FeSe flake with a thickness of 14 nm, comparable to that of sample S1, possesses a superconducting coherence length of 5 nm[44]. Therefore, we conclude that our pristine FeSe flake is in the clean limit.

## Determination of the magnetic phase diagram

The magnetic phase diagram of $Li_xFeSe$ ($0 \leq x \leq 1$) was determined using density-functional theory (DFT) with Hubbard $U$ corrections, as implemented in the Vienna ab initio Simulation Package (VASP)[46,47]. Both monolayer and bulk $Li_xFeSe$ structures were considered to investigate the role of interlayer coupling. Supercells of geometries 2 × 2 × 1 and 2 × 2 × 2 were designed for monolayer and bulk systems, respectively, to sample discrete doping levels $x = 1/4$, 3/8, 1/2, 3/4, and 1 (Supplementary Note 1). The structural models were derived from FeSe in a tetragonal lattice with space group $P4/nmm$, with Li atoms positioned at interstitial sites directly above or below Se ions[15]. For monolayers, a vacuum layer of 15 Å was introduced to eliminate spurious interlayer interactions from periodic boundary conditions. All calculations employed the Perdew-Burke-Ernzerhof (PBE) exchange-correlation functional[48] with a plane-wave energy cutoff of 520 eV. Self-consistent calculations employed Γ-centered $k$-meshes of 11 × 11 × 1 (monolayers) and 9 × 9 × 5 (bulk), ensuring total energy convergence within $10^{-6}$ eV. Structural relaxations were performed for various magnetic configurations using a force convergence criterion of 0.01 eV/Å to determine the structural and magnetic ground state. A Hubbard $U$ was applied to Fe $3d$ orbitals to account for correlation effects, with effective $U$ values[49] systematically varied between 0 and 4 eV to assess the robustness of the magnetic phase diagram.

## Data availability

The data of this study have been deposited in: https://doi.org/10.57760/sciencedb.28024. All other data that support the plots within this paper are available from the corresponding author upon request.

## Code availability

The computer code used for data analysis is available upon request from the corresponding author.

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

## Acknowledgements

We thank Guanghan Cao for helpful discussions. We thank Denis Maryenko, Vadim Grinenko, and Klaus von Klitzing for the critical reading of the manuscript. We thank the WM5 (https://cstr.cn/31125.02.SHMFF.WM5) at the Steady High Magnetic Field Facility, CAS (https://cstr.cn/31125.02.SHMFF), for providing technical support and assistance in data collection and analysis. D. Z. and Q.-K. X. acknowledge financial support from the Ministry of Science and Technology of China (2022YFA1403100); Q.-K. X. acknowledges financial support from National Natural Science Foundation of China (Grant no. 52388201); D. Z. acknowledges financial support from National Natural Science Foundation of China (Grants no. 12361141820, No. T2425009, no. 12274249) and Innovation Program for Quantum Science and Technology (Grant no. 2021ZD0302400); Y. L. acknowledges financial support from

Ministry of Science and Technology of China (Grant no. 2022YFA1403000) and National Natural Science Foundation of China (Grant no. 12274207); Y.-H.W. acknowledges financial support from Shanghai Municipal Science and Technology Major Project (Grant no. 2019SHZDZX01), Ministry of Science and Technology of China (Grant no. 2021YFA1400100) and National Natural Science Foundation of China (Grant no. 12150003); H. C. L. acknowledges financial support from Ministry of Science and Technology of China (Grants no. 2023YFA1406500, No. 2022YFA1403800) and National Natural Science Foundation of China (Grant no. 12274459); J. Z. acknowledges financial support from National Natural Science Foundation of China (Grant no. 12474053).

## Author contributions

D. Z. and Q.-K. X. designed the project. Y. H. fabricated the samples and did the transport measurements with the assistance of J. W. and H. W. Y. H. and F. M. grew the crystals with the guidance of H. L. K. L., Z. L., R. Z., and Y. H. carried out scanning SQUID experiments with the supervision of Y. W. J. C., J. Z., Y. H., J. W., and H. W. carried out the high magnetic field experiments (up to 35 T). J. L. and Y. L. performed the theoretical calculations. D. Z., Y. H. analyzed the data with the input of Q.-K. X. D. Z., Y. H., K. L., Y. L. wrote the paper with the input of H. L., Y. W., and Q.-K. X.

## Competing interests

The authors declare no competing interests.
