## [Transparent Peer Review file · Nature Communications]

Lithium intercalated FeSe as a high-temperature superconducting ferromagnet

Corresponding Author: Professor Ding Zhang

Version 0:

Reviewer comments:

Reviewer #1

(Remarks to the Author)

The authors wrote that one possible route to synthesizing a material with high- T_{sc} is by stacking cuprate or iron-based superconductors with ferromagnetic layers, as in $[(Li,Fe)OH]FeSe$ [12,13].

FeSe becomes a superconductor with $T_{sc} = 43$ K. In practice, studies on Li-intercalated FeSe largely focused on the enhanced superconductivity ($T_{sc} \sim 40$ K) [16-18]. In their compound, itinerant ferromagnetism persists from above 200 K to a temperature well below the superconducting transition temperature (40 K). From this, I conclude that the authors reproduce the 40 K superconductivity in a different way with respect to the literature, and in addition, they just increased the magnetic temperature to 200 Kelvin. It was already known that FeSe is a non-BCS superconductor and this paper does not add further information in this direction.

The paper does not provide anything about the symmetry of the superconductor order parameter. Despite this being an interesting manuscript, I do not think it is enough for publication in Nat. Communication. I think that the paper should be published since there is a lot of work done, as confirmed by the supp. materials, but I suggest publishing the paper in a sister journal as Communication Materials.

Below, there are other comments to motivate my recommendation:

The fact that $T_m \gg T_{sc}$ shows clearly that the two effects are not related.

The author wrote that the anomalous Hall component from the non-trivial Berry curvature stays unchanged throughout the superconducting transition, which confirms that superconductivity does not affect the ferromagnetic state since they are spatially separated. The increase of TC under a magnetic field is less than 1% in the best cases, therefore, it is not a dramatic effect.

They observe the sign change of the AHE and in the region of the sign change there are the humps in the AHE which are not always related to topological Hall effect.

In the abstract, the authors wrote in the abstract that $T_{sc} = 40$ Kelvin, below they wrote that it is 42 Kelvin, while in the plot of Figure 4b T_{sc} seems close to 45 Kelvin.

Reviewer #2

(Remarks to the Author)

Dear Editor,

I have reviewed the manuscript titled:

"Lithium intercalated FeSe as a high-temperature superconducting ferromagnet" by Yi Hu et al.

This work presents the highest superconducting transition temperature T_{sc} of a new ferromagnetic superconductor Li-intercalated FeSe and its electric control. The manuscript demonstrates that T_{sc} is enhanced up to 40 K by applying electric field to the device of FeSe on SiC to intercalate Li ions into FeSe layers while inducing the FM order at $T_m (>225$ K) larger than T_{sc} . The observed reduction of resistance at in-plane magnetic fields indicates the reentrant superconductivity, but the zero resistance T_{sc} does not show it because of broadening of the transition with vortices. Scanning SQUID microscopy and susceptometry observe the coexistence of Meissner screening and FM domains with ~ 2 μm spatial resolution. There is

inhomogeneity in susceptometry but any relationship with FM domains is unclear. DFT+U calculations for monolayer LiFeSe show the FM ground state with over 50% of Li intercalation, consistent with the experimental results. The mechanism of the enhancement of superconductivity and the reentrant superconductivity is yet unclear, but the manuscript successfully demonstrates their great potential for a playground of the interplay between ferromagnetism and superconductivity.

Overall, I find the topic of interest to the field of superconducting spintronics, and the results have potential significance. While some minor corrections and clarifications are needed, I recommend publication after these have been addressed.

Minor Comments

- Figure 2e and Line 188: As the authors said "The susceptometry image at 16 K indicates that nearly the entire sample, within the boundary (demarcated as dashed lines in Fig. 2e), has strong Meissner diamagnetism. ...", the Meissner diamagnetism is inhomogeneous and does not show in some area near the edge. Does these regions remain normal conducting in the lowest T_c state without FM? Does this inhomogeneity originate from FM(Li) density or the pristine sample quality?

- Line 221: In the section "Enhanced superconductivity in an in-plane magnetic field", the authors explain Fig. 3b by assuming two mechanism: (1) reentrant superconductivity and (2) broadening of transition with vortices. But there is not enough discussion with previous studies. For example, there is three possibilities of the reentrant superconductivity: Jaccarino-Peter mechanism with finite magnetic moments(V. Jaccarino and M. Peter: Phys. Rev. Lett. 9, 290 (1962)), Kogan-Nakagawa mechanism in clean two dimensional superconductors(V. G. Kogan & N. Nakagawa, Phys. Rev. B 35, 1700-1707 (1987).) and other low dimensional effects (H. Jeffrey Gardner et al., Nature Physics 7, 895 (2011) DOI: 10.1038/NPHYS2075).

Jaccarino-Peter mechanism may explain this phenomenon. The itinerant electron spins feel opposite effective internal fields by interacting with ordered moments antiferromagnetically (RKKY interactions) and applied fields ferromagnetically (Zeeman effect). At smaller applied fields than the coercive field, the applied field only reduces the effective internal fields at the FM domains along the applied field. At larger applied fields larger than the coercive field, the applied field reduces the effective internal field at the entire sample until the effective internal field becomes zero.

Similar resistance profile has been observed in Pb thin film with Cr magnetic impurity(Nature Physics 7, 895 (2011)).

However Jaccarino-Peter mechanism cannot apply for this case because pristine Pb thin film shows the reentrant superconductivity, indicating another mechanism such as helical phase(V. P. Mineev & K. V. Samokhin, J. Exp. Theor. Phys. 105, 747-763 (1994)).

Jeffrey Gardner et al. also discussed the broadening effect at tilted magnetic field from in-plane magnetic field in the supplementary information, consistent with this manuscript's assumption.

I suggest that adding more discussion is useful for readers.

- Line 156: Should "extracted Hall density (circles)" be (diamond)?

-Line 172: The authors use "MR" in Fig. 1 but $R_{xx}(B)$ in Fig. 2 and call it MR. These confused me in my first reading. I found the definition of MR in the caption of Supplemental figure 11, but I suggest that the definition of "MR" is in the main text or method section.

Sincerely,

Yusuke Iguchi

Geballe Laboratory for Advanced Materials

Stanford University

Version 1:

Reviewer comments:

Reviewer #1

(Remarks to the Author)

I thank the authors for their reply; however, I confirm my previous opinion. The authors show an accurate study on the magnetic part with an increase of Curie temperature and theory for the magnetism; however, there is no solid proof of implications on the superconductivity, but just some speculations.

In my humble opinion, the paper is one step below the standard level of papers published in Nature Communications. I think it should be published in another journal.

Reviewer #2

(Remarks to the Author)

Dear Editor,

In my previous review, I recommended minor revisions. The authors have addressed these points satisfactorily. I support the publication of the revised manuscript.

Sincerely,

Yusuke Iguchi, Ph.D.

Geballe Laboratory for Advanced Materials

Stanford University

Reviewer #1:

The authors wrote that one possible route to synthesizing a material with high- T_{sc} is by stacking cuprate or iron-based superconductors with ferromagnetic layers, as in [(Li,Fe)OH]FeSe [12,13].

FeSe becomes a superconductor with $T_{sc}= 43$ K. In practice, studies on Li-intercalated FeSe largely focused on the enhanced superconductivity ($T_{sc}\sim 40$ K) [16-18]. In their compound, itinerant ferromagnetism persists from above 200 K to a temperature well below the superconducting transition temperature (40 K). From this, I conclude that the authors reproduce the 40 K superconductivity in a different way with respect to the literature, and in addition, they just increased the magnetic temperature to 200 Kelvin. It was already known that FeSe is a non-BCS superconductor and this paper does not add further information in this direction.

[Our reply] We beg to disagree with this opening remark.

First of all, there exists a fundamental distinction between the itinerant ferromagnetism in our system and the ferromagnetism/canted-antiferromagnetism arising from localized magnetic moments. The magnetic layers, such as [(Li,Fe)OH]FeSe, only give rise to the latter type of magnetism. In fact, the weak ferromagnetism of [(Li,Fe)OH]FeSe stems from canted antiferromagnetism. For cuprates, as we include in Fig. 4b, the magnetic layer also shows antiferromagnetic ordering, which can give rise to weak ferromagnetism due to spin canting. By contrast, our system shows clearly ferromagnetic ordering and as the reviewer mentioned it belongs to itinerant ferromagnetism. In order to draw the sharp distinction more clearly, we have revised Fig. 4b by pointing out that the listed cuprates host antiferromagnetic ordering.

Secondly, the enhancement of the ferromagnetic temperature to at least 225 K (the highest temperature we measured) in a superconductor is non-trivial. The weak ferromagnetism in [(Li,Fe)OH]FeSe has an onset temperature as low as 8.5 K and the ferromagnetism in EuFe_2As_2 appears below 20 K. Therefore, we manage to enhance the ferromagnetic temperature of a superconductor by one order of magnitude, which has not been achieved before. We realize that Fig. 4b may give a false impression that the magnetic temperature of our system is only 45 K higher than that of $\text{GdCeRuSr}_2\text{Cu}_2\text{O}_{10}$ (180 K). We have revised Fig. 4b to indicate that T_m in our case corresponds to ferromagnetism while T_m in the listed cuprates represents antiferromagnetism.

With the two clarifications above, we would like to stress that there was no

precedence reporting FeSe as a superconducting ferromagnet, i.e. a system with the coexistence of superconductivity and ferromagnetism. Previous understandings of this material were based on the assumption that the non-BCS superconductivity emerged from a non-magnetic normal state. The well-established understanding of FeSe fails to predict that superconductivity and ferromagnetism can coexist in FeSe. In this regard, the emergence of ferromagnetism in this superconducting compound changes the landscape by opening up a distinct horizon that has never been conceived/explored.

The paper does not provide anything about the symmetry of the superconductor order parameter. Despite this being an interesting manuscript, I do not think it is enough for publication in Nat. Communication. I think that the paper should be published since there is a lot of work done, as confirmed by the supp. materials, but I suggest publishing the paper in a sister journal as Communication Materials.

[Our reply] We would like to thank the reviewer for recognizing the intensive efforts of our work. However, we beg to disagree that our paper does not provide anything about the symmetry of the superconductor order parameter. We observe enhanced superconductivity under an in-plane magnetic field. As we pointed out in the discussion section, such an enhancement suggests spin-triplet pairing, revealing unconventional pairing symmetry of the superconductor order parameter.

In the revised manuscript, we have substantially expanded the discussion on the implication of the enhanced superconductivity. We emphasize more clearly that our results suggest the emergence of a spin-triplet pairing component. Such a possible high-temperature spin-triplet superconductivity is ground breaking since the previous cases only occur at very low temperatures.

Below, there are other comments to motivate my recommendation:

The fact that $T_m \gg T_{sc}$ shows clearly that the two effects are not related.

[Our reply] We are not clear what the reviewer meant by “not related”. Uranium-based superconductors, which show strong interplay between superconductivity and ferromagnetism and are candidates of spin-triplet superconductors, also host $T_m \gg T_{sc}$. The contrast in T_m and T_{sc} can be seen in Fig. R1, in which we show clipped plots from the review paper by D. Aoki, K. Ishida, and J. Flouquet [*J. Phys. Soc. Jpn.* **88**, 022001 (2019)]. Notably, these two compounds show reentrant superconductivity, a clear manifestation of the profound influence of ferromagnetism on superconductivity.

Figure R1 Clips of figures from *J. Phys. Soc. Jpn.* **88**, 022001 (2019). The two plots show the critical magnetic fields as a function of temperature for two Uranium-based superconductors.

Our experiments demonstrate the interplay between superconductivity and ferromagnetism from several aspects: (1) Anomalous Hall effect shows sign reversal around the superconducting transition, indicating the influence of superconductivity on ferromagnetism. (2) Superconductivity gets enhanced in a purely in-plane magnetic field, indicating the influence of ferromagnetism on superconductivity. (3) The magneto-resistance shows down-ward jumps around the coercive field when the temperature is below the onset temperature of superconductivity, whereas it shows up-ward jumps at higher temperatures, reflecting the strong interplay between superconductivity and ferromagnetism.

The author wrote that the anomalous Hall component from the non-trivial Berry curvature stays unchanged throughout the superconducting transition, which confirms that superconductivity does not affect the ferromagnetic state since they are spatially separated. The increase of T_C under a magnetic field is less than 1% in the best cases, therefore, it is not a dramatic effect.

[Our reply] The contribution of anomalous Hall effect from non-trivial Berry curvature stays unchanged because the Berry curvature depends on the band structure, which is not expected to change in the related temperature range of 40 to 50 K because the onset temperature of ferromagnetism is much higher (>225 K). The invariance of band structure is supported by our experimental observation that the Hall density stays nearly constant.

We also beg to differ that the increase of T_{SC} is not a dramatic effect. The effect is prominent because it represents a qualitative change from the conventionally decreasing trend of T_{SC} with B_{\parallel} . It is inappropriate to judge this effect by its quantitative value because a 0% change would correspond to vertically aligned data

points with an extrapolation to infinity—a clearly counterintuitive situation. In order to bring this comparison more clearly, we have revised the corresponding figure to include the reported in-plane magnetic field response of [(Li,Fe)OH]FeSe with a similar T_{sc} . There, only a decreasing trend of T_{sc} with B_{\parallel} was reported. In fact, to the best of our knowledge, the positive shift in T_{sc} with B_{\parallel} has never been reported at such a high temperature. Previous cases, such as Uranium-based superconductors (Fig. R1), showed enhancement but T_{sc} is in the sub-Kelvin regime.

They observe the sign change of the AHE and in the region of the sign change there are the humps in the AHE which are not always related to topological Hall effect.

[Our reply] As we stated in the original text, we suggested that the observed peak/dips were reminiscent to that of topological Hall effect. We are aware that the observation of peak/dips in the Hall effect alone cannot demonstrate the existence of topological Hall effect. This is reflected by our cited references. In particular, refs. 26 and 27 provide alternative scenarios.

To make this point clearer, we have added the following sentence in the corresponding place of the main text: This feature may also reflect the competition between two anomalous Hall components.

In the abstract, the authors wrote in the abstract that $T_{sc}=40$ Kelvin, below they wrote that it is 42 Kelvin, while in the plot of Figure 4b T_{sc} seems close to 45 Kelvin.

[Our reply] The different numbers largely reflect the different definitions of the transition temperatures. In the revised manuscript, we adopt two definitions of the transition temperature: the onset temperature— T_{sc} , and the temperature where the resistance reaches 1% of the normal state resistance— $T_{sc,0}$. The four samples presented in the main text consistently show onset temperatures for the high-temperature superconductivity phase at 45 K. We therefore adopt this temperature as T_{sc} . We use 42 K only when we are referring to $T_{sc,0}$ of sample S1 in state #3 and sample S2 after gating.

Reviewer #2:

This work presents the highest superconducting transition temperature T_{sc} of a new ferromagnetic superconductor Li-intercalated FeSe and its electric control. The manuscript demonstrates that T_{sc} is enhanced up to 40 K by applying electric field to the device of FeSe on SiC to intercalate Li ions into FeSe layers while inducing the FM order at T_m (>225 K) larger than T_{sc} . The observed reduction of resistance at in-plane magnetic fields indicates the reentrant superconductivity, but the zero resistance T_{sc} does not show it because of broadening of the transition with vortices. Scanning SQUID microscopy and susceptometry observe the coexistence of Meissner screening and FM domains with ~ 2 μm spatial resolution. There is inhomogeneity in susceptometry but any relationship with FM domains is unclear. DFT+U calculations for monolayer Li_xFeSe show the FM ground state with over 50% of Li intercalation, consistent with the experimental results. The mechanism of the enhancement of superconductivity and the reentrant superconductivity is yet unclear, but the manuscript successfully demonstrates their great potential for a playground of the interplay between ferromagnetism and superconductivity.

Overall, I find the topic of interest to the field of superconducting spintronics, and the results have potential significance. While some minor corrections and clarifications are needed, I recommend publication after these have been addressed.

[Our reply] We thank Dr. Iguchi for the accurate summary and very positive remark of our work. We are glad to see that our work is recommended for publication after minor revisions.

Minor Comments

- Figure 2e and Line 188: As the authors said “The susceptometry image at 16 K indicates that nearly the entire sample, within the boundary (demarcated as dashed lines in Fig. 2e), has strong Meissner diamagnetism. ...”, the Meissner diamagnetism is inhomogeneous and does not show in some area near the edge. Does these regions remain normal conducting in the lowest T_c state without FM? Does this inhomogeneity originate from FM(Li) density or the pristine sample quality?

[Our reply] Dr. Iguchi is correct that the Meissner diamagnetism does not cover uniformly the complete sample at 16 K. By comparing to the magnetometry image, we see that there exist two types of regions near the boundary of the sample: (A) the top corner of the sample shows no diamagnetism signal and no DC flux signal. This type of areas seems to be in the low- T_{sc} state with the absence of both diamagnetism and

ferromagnetism; (B) the lower right part of the sample close to the boundary shows no diamagnetism but positive flux signal seems to extend into this region. It is consistent with the behavior of a region with slightly higher lithium distribution (>50%) such that superconductivity is fully suppressed but ferromagnetism persists. We speculate that this inhomogeneity mainly stems from the inhomogeneous distribution of lithium close to the boundary of the sample.

- Line 221: In the section “Enhanced superconductivity in an in-plane magnetic field”, the authors explain Fig. 3b by assuming two mechanisms: (1) reentrant superconductivity and (2) broadening of transition with vortices. But there is not enough discussion with previous studies. For example, there are three possibilities of the reentrant superconductivity: Jaccarino-Peter mechanism with finite magnetic moments (V. Jaccarino and M. Peter: Phys. Rev. Lett. 9, 290 (1962)), Kogan-Nakagawa mechanism in clean two dimensional superconductors (V. G. Kogan & N. Nakagawa, Phys. Rev. B 35, 1700-1707 (1987).) and other low dimensional effects (H. Jeffrey Gardner et al., Nature Physics 7, 895 (2011) DOI: 10.1038/NPHYS2075).

Jaccarino-Peter mechanism may explain this phenomenon. The itinerant electron spins feel opposite effective internal fields by interacting with ordered moments antiferromagnetically (RKKY interactions) and applied fields ferromagnetically (Zeeman effect). At smaller applied fields than the coercive field, the applied field only reduces the effective internal fields at the FM domains along the applied field. At larger applied fields larger than the coercive field, the applied field reduces the effective internal field at the entire sample until the effective internal field becomes zero.

Similar resistance profile has been observed in Pb thin film with Cr magnetic impurity (Nature Physics 7, 895 (2011)). However Jaccarino-Peter mechanism cannot apply for this case because pristine Pb thin film shows the reentrant superconductivity, indicating another mechanism such as helical phase (V. P. Mineev & K. V. Samokhin, J. Exp. Theor. Phys. 105, 747-763 (1994)).

Jeffrey Gardner et al. also discussed the broadening effect at tilted magnetic field from in-plane magnetic field in the supplementary information, consistent with this manuscript’s assumption.

I suggest that adding more discussion is useful for readers.

[Our reply] This is a very detailed and helpful comment. We especially thank Dr. Iguchi for pointing out the broadening effect of the previous work, which helps explain our observation. We have added a sentence: “Such a broadening effect in addition to the enhanced superconductivity was reported in the study of Pb films under a tiny tilting

Figure R2 Clips of figures from the listed two references. Figures show typical reentrant superconductivity induced by high magnetic fields.

of the magnetic field.” in the section discussing Fig. 3b. We have substantially revised our discussions by fully taking into account the three mechanisms pointed out by Dr. Iguchi. Our analysis indicates that the three mechanisms seem unable to fully explain the experimental results. We thus speculate on the possible involvement of spin-triplet pairing as the origin for the enhanced superconductivity. In general, we acknowledge that our work gets significantly improved by making an expanded discussion requested by Dr. Iguchi.

On the Jaccarino-Peter effect, we would like to point out that it manifests in real systems as two superconducting phases: 1. the conventional phase in which superconductivity gets suppressed with the applied magnetic field; 2. the reentrant phase which occurs at sufficiently high magnetic fields that compensates for the internal field. For convenience, we show in Fig. R2 the typical data demonstrating the reentrant superconductivity. The authors in the corresponding references attributed the reentrant phase to the Jaccarino-Peter effect. We reason that the usual suppression of superconductivity at low magnetic field occurs in the scenario of Jaccarino and Peter because of the alignment of magnetic ion spins, as we schematically illustrate in Fig. R3a (middle inset). In comparison, our system seems to show only a monotonic enhancement of superconductivity with the in-plane magnetic field. We show this behavior in Fig. R3b, where we sweep the magnetic field continuously in the superconducting transition regime of sample S2.

Figure R3 a. Schematic drawing of the expected magnetic field dependence of resistance governed by the Jaccarino-Peter (J-P) effect. Gray/blue circles represent the magnetic ions/electrons. Arrows indicate the spin direction. **b.** Experimentally measured resistance of sample S2 in the superconducting transition regime as a function of the magnetic field. Angles indicate the orientation of the magnetic field relative to the sample plane (inset).

- Line 156: Should “extracted Hall density (circles)” be (diamond)?

[Our reply] We thank Dr. Iguchi for the careful reading of our manuscript. We have revised the texts correspondingly.

-Line 172: The authors use “MR” in Fig. 1 but $R_{xx}(B)$ in Fig. 2 and call it MR. These confused me in my first reading. I found the definition of MR in the caption of Supplemental figure 11, but I suggest that the definition of “MR” is in the main text or method section.

[Our reply] Following this nice suggestion, we have added the definition of MR in the main text. We have also revised the descriptions of Fig. 2 by referring to the behavior as the magnetic field dependence of the longitudinal resistance $R_{xx}(B)$.